# Taking a Step Back with KCal: Multi-Class Kernel-Based Calibration for Deep Neural Networks

**Zhen Lin**[1]     **Shubhendu Trivedi**     **Jimeng Sun**[1,2]
[1] Department of Computer Science, University of Illinois at Urbana-Champaign
[2] Carle Illinois College of Medicine, University of Illinois at Urbana-Champaign
`{zhenlin4,jimeng}@illinois.edu   shubhendu@csail.mit.edu`

## Abstract

Deep neural network (DNN) classifiers are often overconfident, producing miscalibrated class probabilities. In high-risk applications like healthcare, practitioners require *fully calibrated* probability predictions for decision-making. That is, conditioned on the prediction *vector*, *every* class' probability should be close to the predicted value. Most existing calibration methods either lack theoretical guarantees for producing calibrated outputs, reduce classification accuracy in the process, or only calibrate the predicted class. This paper proposes a new Kernel-based calibration method called KCal. Unlike existing calibration procedures, KCal does not operate directly on the logits or softmax outputs of the DNN. Instead, KCal learns a metric space on the penultimate-layer latent embedding and generates predictions using kernel density estimates on a calibration set. We first analyze KCal theoretically, showing that it enjoys a provable *full* calibration guarantee. Then, through extensive experiments across a variety of datasets, we show that KCal consistently outperforms baselines as measured by the calibration error and by proper scoring rules like the Brier Score. Our code is available at `https://github.com/zlin7/KCal`.

## 1 Introduction

The notable successes of Deep Neural Networks (DNNs) in complex classification tasks, such as object detection (Ouyang & Wang, 2013), speech recognition (Deng et al., 2013), and medical diagnosis (Qiao et al., 2020; Biswal et al., 2017), have made them essential ingredients within various critical decision-making pipelines. In addition to the classification accuracy, a classifier should ideally also generate reliable uncertainty estimates represented in the predicted probability vector. An influential study (Guo et al., 2017) reported that modern DNNs are often overconfident or *miscalibrated*, which could lead to severe consequences in high-stakes applications such as healthcare (Jiang et al., 2012).

Calibration is the process of closing the gap between the prediction and the ground truth distribution given this prediction. For a $K$-classification problem, with covariates $X \in \mathcal{X}$ and the label $Y \in \mathcal{Y} = [K]$, denote our classifier $\mathcal{X} \mapsto \Delta^{K-1}$ as $\hat{\mathbf{p}} = [\hat{p}_1, \ldots, \hat{p}_K]$, with $\Delta^{K-1}$ being ($K$-1)-simplex. Then,

**Definition 1.** *(Full Calibration (Vaicenavicius et al., 2019))* $\hat{\mathbf{p}}$ *is fully-calibrated if* $\forall k \in [K]$:

$$\forall \mathbf{q} = [q_1, \ldots, q_K] \in \Delta^{K-1}, \mathbb{P}\{Y = k | \hat{\mathbf{p}}(X) = \mathbf{q}\} = q_k. \tag{1}$$

It is worth noting that Def. (1) implies *nothing about accuracy*. In fact, ignoring $X$ and simply predicting $\pi$, the class frequency vector, results in a fully calibrated but inaccurate classifier. As a result, our goal is always to improve calibration *while maintaining accuracy*. Another important requirement is that $\hat{\mathbf{p}} \in \Delta^{K-1}$. Many binary calibration methods such as Zadrozny & Elkan (2001; 2002) result in vectors that are not interpretable as probabilities, and have to be normalized.

Many existing works only consider *confidence calibration* (Guo et al., 2017; Zhang et al., 2020; Wenger et al., 2020; Ma & Blaschko, 2021), a much weaker notion than that encapsulated by Def. (1) and only calibrates the predicted class (Kull et al., 2019; Vaicenavicius et al., 2019).

**Definition 2.** *(Confidence Calibration)* $\hat{\mathbf{p}}$ *is confidence-calibrated if:*

$$\forall q \in [0,1], \mathbb{P}\{Y = \arg\max_k \hat{p}_k(X) | \max_k \hat{p}_k(X) = q\} = q. \tag{2}$$

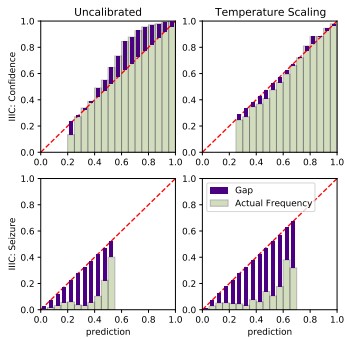

Figure 1: Reliability diagrams for confidence calibration (top) and Seizure (bottom). The popular temperature scaling (right) only calibrates the confidence, leaving Seizure poorly calibrated. See Figure 2 and the Appendix for complete reliability diagrams.

However, confidence calibration is far from sufficient. Doctors need to perform differential diagnoses on a patient, where multiple possible diseases should be considered with proper probabilities for all of them, not only the most likely diagnosis. Figure 1 shows an example where the confidence is calibrated, but prediction for important classes like Seizure is poorly calibrated. A classifier can be confidence-calibrated but not useful for such tasks if the probabilities assigned to most diseases are inaccurate.

Recent research effort has started to focus on full calibration, for example, in Vaicenavicius et al. (2019); Kull et al. (2019); Widmann et al. (2019). We approach this problem by leveraging the latent neural network embedding in a nonparametric manner. Nonparametric methods such as histogram binning (HB) (Zadrozny & Elkan, 2001) and isotonic regression (IR) (Zadrozny & Elkan, 2002), are natural for calibration and have become popular. Gupta & Ramdas (2021) recently showed a calibration guarantee for HB. However, HB usually leads to noticeable drops in accuracy (Patel et al., 2021), and IR is prone to overfitting (Niculescu-Mizil & Caruana, 2005). Unlike existing methods, we take one step back and train a new low-dimensional metric space on the penultimate-layer embeddings of DNNs. Then, we use a kernel density estimation-based classifier to predict the class probabilities directly. We refer to our **K**ernel-based **Cal**ibration method as KCal. Unlike most calibration methods, KCal provides high probability error bounds for *full calibration* under standard assumptions. Empirically, we show that with little overhead, KCal outperforms all existing calibration methods in terms of calibration quality, across multiple tasks and DNN architectures, while maintaining and sometimes improving the classification accuracy.

**Summary of Contributions**:

- We propose KCal, a principled method that calibrates DNNs using kernel density estimation on the *latent embeddings*.
- We present an efficient pipeline to train KCal, including a dimension-reducing projection and a stratified sampling method to facilitate efficient training.
- We provide finite sample bounds for the calibration error of KCal-calibrated output under standard assumptions. To the best of our knowledge, this is the first method with a full calibration guarantee, especially for neural networks.
- In extensive experiments on multiple datasets and state-of-the-art models, we found that KCal outperforms existing calibration methods in commonly used evaluation metrics. We also show that KCal provides more reliable predictions for important classes in the healthcare datasets.

The code to replicate all our experimental results is submitted along with supplementary materials.

## 2 RELATED WORK

Research on calibration originated in the context of meteorology and weather forecasting (see Murphy & Winkler (1984) for an overview) and has a long history, much older than the field of machine learning (Brier, 1950; Murphy & Winkler, 1977; Degroot & Fienberg, 1983). We refer to Filho et al. (2021) for a holistic overview and focus below on methods proposed in the context of modern neural networks. Based on underlying methodological similarities, we cluster them into distinct categories.

**Scaling:** A popular family of calibration methods is based on scaling, in which a mapping is learned from the predicted logits to probability vectors. Confidence calibration scaling methods include temperature scaling (TS) (Guo et al., 2017) and its antecedent Platt scaling (Platt, 1999), an ensemble of TS (Zhang et al., 2020), Gaussian-Process scaling (Wenger et al., 2020), combining a base calibrator (TS) with a rejection option (Ma & Blaschko, 2021). Matrix scaling with regularization

was also used to perform full calibration (Kull et al., 2019). While some scaling-based methods can be data-efficient, there are no known theoretical guarantees for them to the best of our knowledge.

**Binning:** Another cluster of solutions relies on binning and its variants, and includes uniform-mass binning (Zadrozny & Elkan, 2001), scaling before binning (Kumar et al., 2019), and mutual-information-maximization-based binning (Patel et al., 2021). Isotonic regression (Zadrozny & Elkan, 2002) is also often interpreted as binning. Uniform-mass binning (Zadrozny & Elkan, 2001) has a distribution-free finite sample calibration guarantee (Gupta & Ramdas, 2021) and asymptotic convergent ECE estimation (Vaicenavicius et al., 2019). However, in practice, binning tends to decrease accuracy (Patel et al., 2021; Guo et al., 2017). Binning can also be considered a member of the broader nonparametric calibration family of methods. Such methods also include Gaussian Process Calibration (Wenger et al., 2020), which however also only considers confidence calibration.

**Loss regularization:** There are also attempts to train a calibrated DNN to begin with. Such methods typically add a suitable regularizer to the loss function (Karandikar et al., 2021; Mukhoti et al., 2020; Kumar et al., 2018), which can sometimes result in expensive optimization and reduction in accuracy.

**Use of Kernels:** Although not directly used for calibration, kernels have also been used for uncertainty quantification for deep learning classification. In classification with rejection, the k-nearest-neighbors algorithm (kNN), closely related to kernel-based methods, has been used to provide a "confidence measure" which is used to make a binary decision (i.e., whether to reject or to predict) (Papernot & McDaniel, 2018; Jiang et al., 2018). Recently, continuous kernels have also been used to measure calibration quality or used as regularization during training (Widmann et al., 2019; Kumar et al., 2018). Zhang et al. (2020) introduced a kernel density estimation (KDE) proxy estimator for estimating ECE. However, it uses a un-optimized kernel over $\Delta^{K-1}$, and shows the KDE-ECE estimator (but not the calibration map) is consistent. To the best of our knowledge, use of trained KDE to calibrate predictions hasn't been proposed before. Further, we also provide a bound on the calibration error.

## 3 KCAL: KERNEL-BASED CALIBRATION

In this section, we formally introduce KCal, study its calibration properties theoretically, and present crucial implementation details and comparisons with other methods. Specifically, in Section 3.1, we discuss how to construct (automatically) calibrated predictions for test data using a calibration set $\mathcal{S}_{\mathrm{cal}}$. Doing so requires a well-trained kernel and metric space, and we describe a procedure to train such a kernel in Section 3.2. In Section 3.3, we show that an appropriate shrinkage rate of the bandwidth ensures that the KCal prediction is automatically calibrated. Sections 3.4 provides implementation details. Finally, in Section 3.5, we compare and contrast KCal with existing methods.

### 3.1 CLASSIFICATION WITH KERNEL DENSITY ESTIMATION

Following the calibration literature, we first require a holdout calibration set $\mathcal{S}_{\mathrm{cal}} = \{X_i, Y_i\}_{i=1}^{N}$. In KCal, we fix a kernel function $\hat{\phi}$ which is learned (the learning procedure is described in Section 3.2). For a new datum $X_{N+1}$, the class probability $\hat{\mathbf{p}}_k(X_{N+1})$ takes the following form:

$$\hat{\mathbf{p}}_k(X_{N+1}; \hat{\phi}, \mathcal{S}_{\mathrm{cal}}) = \frac{\sum_{(x,y)\in\mathcal{S}_{\mathrm{cal}}^k} \hat{\phi}(x, X_{N+1})}{\sum_{(x,y)\in\mathcal{S}_{\mathrm{cal}}} \hat{\phi}(x, X_{N+1})}, \tag{3}$$

where $\mathcal{S}_{\mathrm{cal}}^k := \{(x, y) \in \mathcal{S}_{\mathrm{cal}} | y = k\}$. The notation $\hat{\mathbf{p}}_k(X_{N+1}; \hat{\phi}, \mathcal{S}_{\mathrm{cal}})$ emphasizes the dependence on $\hat{\phi}$ and $\mathcal{S}_{\mathrm{cal}}$. However, we will use $\hat{\mathbf{p}}_k(X_{N+1})$ when the dependence is clear from context.

**Remarks**: What we have described is essentially the classical nonparametric procedure of applying kernel density estimation for classification. For a moment, suppose we know the true density function $f_k$ of $\mathcal{P}_k$ (the distribution of all the data in class $k$), and the proportion of class $k$, denoted $\pi_k$ (such that $\sum_{k\in[K]} \pi_k = 1$). Then, for any particular $x_0$, using the Bayes rule we get:

$$\mathbb{P}\{Y = k | X = x_0\} = \frac{f_k(x_0)\pi_k}{\sum_{k'\in[K]} f_{k'}(x_0)\pi_{k'}}. \tag{4}$$

Now, replacing $f_k$ with the kernel density estimate $\hat{f}_k(x_0) := (\sum_{(x,y)\in\mathcal{S}_{\mathrm{cal}}^k} \hat{\phi}_b(x, x_0))/|\mathcal{S}_{\mathrm{cal}}^k|$, and the class proportion $\pi_k$ with $\hat{\pi}_k := |\mathcal{S}_{\mathrm{cal}}^k|/|\mathcal{S}_{\mathrm{cal}}|$ we get back Eq. (3).

## 3.2 TRAINING

For good performance under the kernel density framework, it is crucial to employ an appropriate kernel function $\hat{\phi}$, which in turn relies on the choice of the underlying metric. Therefore, we train a metric space on top of the penultimate layer embeddings of deep learning models.

To begin, we assume a deep neural network is already trained on $\mathcal{S}_{\text{train}} = \{X_i^{train}, Y_i^{train}\}_{i=1}^M$. We place no limitations on the form of loss function, optimizer, or the model architecture. However, we do require the neural net to compute an embedding before a final prediction layer, which is always the case in modern classification models. We denote the embedding function from $\mathcal{X} \mapsto \mathbb{R}^h$ as $\mathbf{f}$.

Given a base "mother kernel" function $\phi$, such as the Radial Basis Function (RBF) kernel, we denote the kernel with bandwidth $b$ as $\phi_b := \frac{1}{b}\phi(\frac{\cdot}{b})$. We parameterize the learnable kernel as:

$$\hat{\phi}(x, x') := \hat{\phi}_{\mathbf{\Pi}, \mathbf{f}, b}(x, x') := \phi_b(\mathbf{\Pi}(\mathbf{f}(x)) - \mathbf{\Pi}(\mathbf{f}(x'))). \tag{5}$$

where $\mathbf{\Pi} : \mathbb{R}^h \mapsto \mathbb{R}^d$ is a dimension-reducing projection parameterized by a shallow MLP (Section 3.4). Since the inference time is linear in $d$, letting $d < h$ also affords computational benefits.

Given that the embedding function $\mathbf{f}(x)$ from the neural network is fixed, the only learnable entities are $b$ and $\mathbf{\Pi}$. In the training phase, we fix $b = 1$, and train $\mathbf{\Pi}$ using (stochastic) gradient descent and log-loss. The specific value of $b$ does not matter since it can be folded into $\mathbf{\Pi}$. Let us denote $\mathcal{S}_{\text{train}}^k = \{(x, y) \in \mathcal{S}_{\text{train}} : y = k\}$. In each iteration, we randomly sample two batches of data from $\mathcal{S}_{\text{train}}$ - the prediction data, denoted as $\mathcal{S}_{\text{train}}^B$, to evaluate $\mathbf{\Pi}$, and "background" data for each $k$, denoted as $\mathcal{B}^k$, from $\mathcal{S}_{\text{train}}^k \setminus \mathcal{S}_{\text{train}}^B$ to construct the KDE classifier. Then, the prediction for any $x_j$ is given by

$$\hat{\mathbf{p}}_k(x_j; \hat{\phi}, \mathcal{S}_{\text{train}} \setminus \mathcal{S}_{\text{train}}^B) := \left( \sum_{(x,y) \in \mathcal{B}^k} \frac{|\mathcal{S}_{\text{train}}^k \setminus \mathcal{S}_{\text{train}}^B|}{|\mathcal{B}^k|} \hat{\phi}(x, x_j) \right) / \left( \sum_{k', (x,y) \in \mathcal{B}^{k'}} \frac{|\mathcal{S}_{\text{train}}^{k'} \setminus \mathcal{S}_{\text{train}}^B|}{|\mathcal{B}^{k'}|} \hat{\phi}(x, x_j) \right) \tag{6}$$

where $\hat{\phi}$ is shorthand for $\hat{\phi}_{\mathbf{\Pi}, \mathbf{f}, b=1}$ defined in Eq. (5).

---

**Algorithm 1** Overview of KCal

**Input**:
$\mathcal{S}_{\text{train}}$: $\{(X_i^{train}, Y_i^{train})\}_{i=1}^M$ used to train the NN
$\mathcal{S}_{\text{cal}}$: $\{(X_i, Y_i)\}_{i=1}^N$ calibration set
$\mathbf{f}$ : Embedding function $\mathcal{X} \to \mathbb{R}^h$ (trained NN)
$X_{N+1}$: Unseen datum for prediction
**Training (of the projection $\mathbf{\Pi}$)**:
    Denote $\mathcal{S}_{\text{train}}^k := \{(x, y) \in \mathcal{S}_{\text{train}} | y = k\}$.
    Denote $\phi_b$ as a base kernel function (e.g. RBF) with bandwidth $b$.
    **repeat**
        Sample $\mathcal{S}_{\text{train}}^B = \{(x_j, y_j)\}_{j=1}^B$ from $\mathcal{S}_{\text{train}}$.
        Compute $\hat{\mathbf{p}}(x_j)$ via Eq. (6).
        Loss $l \leftarrow \frac{1}{B} \sum_{j=1}^B LogLoss(\hat{\mathbf{p}}(x_j), y_j)$.
        Update $\mathbf{\Pi}$ with (stochastic) gradient descent.
    **until** the loss $l$ does not improve.
    Set $\hat{\phi}_b \leftarrow \hat{\phi}_{\mathbf{\Pi}, \mathbf{f}, b}$ for inference.
**Inference**:
    Denote $\mathcal{S}_{\text{cal}}^k := \{(x, y) \in \mathcal{S}_{\text{cal}} | y = k\}$.
    Tune $b^*$ on $\mathcal{S}_{\text{cal}}$ by minimizing log loss.
    $\hat{\mathbf{p}}_k(X_{N+1}) \leftarrow \frac{\sum_{(x,y) \in \mathcal{S}_{\text{cal}}^k} \hat{\phi}_{b^*}(x, X_{N+1})}{\sum_{(x,y) \in \mathcal{S}_{\text{cal}}} \hat{\phi}_{b^*}(x, X_{N+1})}$.

---

The log-loss is given formally by

$$L = -\frac{1}{B} \sum_{(x,y) \in \mathcal{S}_{\text{train}}^B} \log \hat{\mathbf{p}}_y(x; \hat{\phi}, \mathcal{S}_{\text{train}} \setminus \mathcal{S}_{\text{train}}^B).$$

Finally, we pick a $b = b^*$ on the calibration set $\mathcal{S}_{\text{cal}}$ using cross-validation. This is because $b$ should be chosen contingent on the sample size (Section 3.3). Choosing $b$ can be done efficiently (Section 3.4). Algorithm 1 summarizes the steps we explicated upon so far.

## 3.3 THEORETICAL ANALYSIS: CALIBRATION COMES FREE

In the previous section, we have only described a procedure to improve the prediction accuracy for $\hat{\mathbf{p}}$ on $\mathcal{S}_{\text{train}}$. This section will show that calibration comes free with the $\hat{\mathbf{p}}$ obtained using Algorithm 1. In particular, we show that as the sample-size for each class in $\mathcal{S}_{\text{cal}}$ increases, $\hat{\mathbf{p}}$ converges to the true frequency vector of $Y$ given the prediction. For smoother presentation, we only state the relevant claims in what follows. Detailed proofs are presented in the Appendix.

To begin, we make a few standard assumptions, such as in Chacón & Duong (2018), including:

- ($\forall k$) The density on the embedded space, $\mathbf{\Pi}(\mathbf{f}(\mathcal{X} | Y = k))$, denoted as $f_{\mathbf{\Pi} \circ \mathbf{f}, k}$, is square integrable and twice differentiable, with all second order partials bounded, continuous, and square integrable.
- $\phi$ is spherically symmetric, with a finite second moment.

Lemma 3.1 and 3.2 focus on an arbitrary class $k$ and ignore the subscript $k$ to the density $f$ for readability. We denote the size $|\mathcal{S}_{\text{cal}}^k| = m$. Intuitively, due to the bias-variance trade-off, a suitable bandwidth $b$ will depend on $m$: A small $b$ reduces bias, but with the finite $m$, a smaller $b$ also leads to increased variance. Thus, $b$ should go to 0 "slowly", which is formally stated below:

**Lemma 3.1.** *For almost all $x$, if $b^d m \to \infty$ and $b \to 0$ as $m \to \infty$, then we have*

$$\|\hat{f}_{\boldsymbol{\Pi} \circ \mathbf{f}, k}(x) - f_{\boldsymbol{\Pi} \circ \mathbf{f}, k}(x)\|_2 \xrightarrow{P} 0 \text{ as } m \to \infty. \tag{7}$$

Here $\hat{f}_{\boldsymbol{\Pi} \circ \mathbf{f}, k}$ is the estimated $f_{\boldsymbol{\Pi} \circ \mathbf{f}, k}$ using $\mathcal{S}_{\text{cal}}$. Recall that $d$ is the dimension of $\boldsymbol{\Pi}(\mathbf{f}(\mathcal{X}))$. We will call such a bandwidth $b$ *admissible*, and we sometimes write $b(m)$ to emphasize the dependence on $m$. The following lemma gives the optimal admissible bandwidth:

**Lemma 3.2.** *The optimal bandwidth is $b = \Theta(m^{-\frac{1}{d+4}})$, which leads to the fastest decreasing MSE (i.e. $\mathbb{E}[\|\hat{f}_{\boldsymbol{\Pi} \circ \mathbf{f}, k}(x) - f_{\boldsymbol{\Pi} \circ \mathbf{f}, k}(x)\|_2]$) of $O(m^{-\frac{4}{d+4}})$.*

Now we are in a position to present the main theoretical results. In the following, $m$ denotes the rarest class's count ($m := \min_k \{|\mathcal{S}_{\text{cal}}^k|\}$. Theorem 3.3 provides a bound between $\hat{\mathbf{p}}$ and the true conditional probability vector on the embedded space $\mathbf{p}(\boldsymbol{\Pi}(\mathbf{f}(X)))$:

**Theorem 3.3.** *Fixing $x$ such that the density of $\boldsymbol{\Pi}(\mathbf{f}(x))$ is positive, with $b(m) = \Theta(m^{-\frac{1}{d+4}})$, for any $\lambda \in (0, 2)$:*

$$\mathbb{P}\{|\hat{\mathbf{p}}_k(x) - \mathbf{p}_k(\boldsymbol{\Pi}(\mathbf{f}(x)))| > (3K+1)Cm^{\frac{-\lambda}{d+4}}\} \leq Ke^{-Bm^{\frac{4-2\lambda}{d+4}}} \tag{8}$$

$$\text{where } \mathbf{p}_k(\boldsymbol{\Pi}(\mathbf{f}(x))) := \mathbb{P}\{Y = k | \boldsymbol{\Pi}(\mathbf{f}(X)) = \boldsymbol{\Pi}(\mathbf{f}(x))\} \tag{9}$$

*for some constant $B$ and $C$. As a corollary, $\hat{\mathbf{p}}(x) \xrightarrow{almost\ surely} \mathbf{p}(\boldsymbol{\Pi}(\mathbf{f}(x)))$ as $m \to \infty$.*

Next, we bound the full calibration error with additional standard assumptions. More specifically, we use and build upon the main uniform convergence result for classical KDE presented in Jiang (2017), to obtain Theorem 3.4:

**Theorem 3.4.** *Assume $f_{\boldsymbol{\Pi} \circ \mathbf{f}, k}$ is $\alpha$-Hölder continuous and bounded away from 0 for any $k$. For an admissible $b(m)$ with shrinkage rate $\Theta((\frac{\log m}{m})^{\frac{1}{d+2\alpha}})$, for some constants $B$ and $C$ we have:*

$$\mathbb{P}\{\sup_{X,k} |\hat{\mathbf{p}}_k(X) - \mathbb{P}\{Y = k | \hat{\mathbf{p}}(X)\}| > (3K+1)C(\frac{\log m}{m})^{\frac{\alpha}{d+2\alpha}}\} \leq K(m^{-1} + m^{-B\frac{2\alpha}{d+2\alpha}} m^{\frac{d}{d+2\alpha}}). \tag{10}$$

We now proceed to present details pertaining to the efficient implementation of KCal.

### 3.4 IMPLEMENTATION TECHNIQUES

**Efficient Training**: As might be immediately apparent, utilizing algorithm 1 for prediction using full $\mathcal{S}_{\text{train}} \setminus \mathcal{S}_{\text{train}}^B$ can be an expensive exercise. In order to afford a training speedup, we consider a random subset from $\mathcal{S}_{\text{train}} \setminus \mathcal{S}_{\text{train}}^B$ using a modified stratified sampling. Specifically, we take $m$ random samples from each $\mathcal{S}_{\text{train}}^k$, denoted as $\mathcal{S}_{\text{train}}^{k,m}$, and replace the right-hand side of Eq. 6 with:

$$\Big( \sum_{(x,y) \in \mathcal{S}_{\text{train}}^{k,m}} \frac{|\mathcal{S}_{\text{train}}^k|}{m} \hat{\phi}(x, x_0) \Big) / \Big( \sum_{k' \in [K], (x,y) \in \mathcal{S}_{\text{train}}^{k',m}} \frac{|\mathcal{S}_{\text{train}}^{k'}|}{m} \hat{\phi}(x, x_0) \Big). \tag{11}$$

The re-scaling term $\frac{|\mathcal{S}_{\text{train}}^k|}{m}$ is crucial to get an unbiased estimate of $\hat{f}_k \hat{\pi}_k$. The stratification employed makes the training more stable, while also reducing the estimation variance for rarer classes (more details in Appendix B). The overall complexity is now $O(KmdhB)$ per batch. In all experiments, we used $m = 20$ and $B = 64$.

**Form of $\boldsymbol{\Pi}$**: While there is considerable freedom in choosing a suitable form for $\boldsymbol{\Pi}$, we parameterize $\boldsymbol{\Pi}$ with a two layer MLP with a skip connection. Consequently, $\boldsymbol{\Pi}$ can reduce to linear projection when sufficient, and be more expressive when necessary. We also experimented with using only a linear projection, the results for which are included in the appendix. We fix the output dimension to $d = \min\{dim(\mathbf{f}), 32\}$, except for ImageNet ($d = 128$).

**Bandwidth Selection**: Finally, to find the optimal bandwidth using $\mathcal{S}_{\text{cal}}$, we use Golden-Section search (Kiefer, 1953) to find the log-loss-minimizing $b^*$. This takes $O(\log \frac{ub - lb}{tol})$ steps where $[lb, ub]$ is the search space, and $tol$ is the tolerance. Essentially, we assume that the loss is a convex function with respect to $b$, permitting an efficient search (see Appendix H, which presents empirical evidence that the convexity assumption is valid across datasets).

### 3.5 COMPARISONS WITH EXISTING CALIBRATION METHODS

Most existing calibration methods discussed in Section 2 and KCal all utilize a holdout calibration set. However, unlike KCal, existing works usually fix the last neural network layer. KCal, on the other hand, "takes a step back", and replaces the last prediction layer with a kernel density estimation based classifier. Since the DNN $\mathbf{f}$ is fixed regardless of whether we use the original last layer or not, we are really comparing a KDE classifier (KCal) with linear models trained in various ways, after mapping all the data with $\mathbf{f}$. Note that this characterization is true for most existing methods, with a few exceptions (e.g., those summarized under "loss regularization" in Section 2).

Employing a KDE classifier affords some clear advantages such as a straightforward convergence guarantee and some interpretability[1]. Furthermore, KCal can also be improved in an online fashion, a benefit especially desirable in certain high-stakes applications such as in healthcare. For example, a hospital can calibrate a trained model prior to deployment using its own patient data (which is usually not available to train the original model) as it becomes available.

Another important advantage of KCal is concerning normalization. In fact, simultaneously calibrating all classes while satisfying the constraint that $\hat{\mathbf{p}} \in \Delta^{K-1}$ is a distinguishing challenge for multi-class calibration. Many calibration methods perform one-vs-rest calibration for each class, and require a separate normalization step at test time (Zadrozny & Elkan, 2001; 2002; Patel et al., 2021; Gupta et al., 2021). This creates a gap between training and testing and could lead to drastic drop in performance (Section 4). On the other hand, KCal automatically satisfies $\hat{\mathbf{p}} \in \Delta^{K-1}$, and the normalization is consistent during training and testing.

A disadvantage of KCal is the need to remember the $\mathbf{\Pi}(\mathbf{f}(\mathcal{S}_{\text{cal}}))$ used to generate the KDE prediction. This is however mitigated to a large extent by the dimension reduction step, which already reduces the computational overhead significantly[2]. For example, in one of our experiments on CIFAR-100, there are 160K (5K images, $d = 32$) scalars to remember, which is only 0.2% of the parameters (85M+) of the original DNN (ViT-base-patch16). Moreover, KDE inference is trivial to parallelize on GPUs. There is also a rich, under-explored, literature to further speed up the inference. Examples include, KDE merging (Sodkomkham et al., 2016), Dual-Tree (Gray & Moore, 2003), and Kernel Herding (Chen et al., 2010). These methods can easily be used in conjunction with KCal.

## 4 EXPERIMENTS

### 4.1 DATA AND NEURAL NETWORKS

We utilize two sets of data: computer vision benchmarks on which previous calibration methods were tested, and health monitoring datasets where *full* calibration is crucial for diagnostic applications. Table 1 summarizes the datasets and their splits.

Table 1: Dataset summary: Splits and number of classes ($K$).

| Dataset | IIIC | IIIC(pat) | ISRUC | ISRUC(pat) | PN2017 | C10 | C100 | SVHN | ImageNet |
|---|---|---|---|---|---|---|---|---|---|
| Train | 103,818 | 1,936 | 61,841 | 69 | 15,087 | 45,000, | 45,000 | 65,931 | 1,281,167 |
| Calibration | 1,787 | 77 | 1,372 | 6 | 253 | 5,000 | 5,000 | 7,326 | 25,000 |
| Test | 33,953 | 684 | 26,070 | 24 | 4,813 | 10,000 | 10,000 | 26,032 | 25,000 |
| $K$ | 6 | 6 | 5 | 5 | 4 | 10 | 100 | 10 | 1,000 |

**Benchmark data** Following Kull et al. (2019), we use multiple image benchmark datasets, including CIFAR-10, CIFAR-100, and SVHN (Krizhevsky, 2009; Netzer et al., 2011). We reserve 10% of the training data as the calibration set. We fine-tune pretrained ViT (Dosovitskiy et al., 2021) and MLP-Mixer (Mixer) (Tolstikhin et al., 2021) from the `timm` library (Wightman, 2019). We chose ViT and Mixer because they are the state-of-the-art neural architectures in computer vision, and accuracy should come before calibration quality. We also included the ImageNet dataset (Deng et al., 2009) and use the pretrained Inception ResNet V2 (Szegedy et al., 2017) following Patel et al. (2021).

**Health monitoring data** We also use three health monitoring datasets for diagnostic tasks: IIIC (Jing et al., 2021), an ictal-interictal-injury-continuum (IIIC) patterns classification dataset; ISRUC (Khalighi et al., 2016), a sleep staging (classification) dataset using polysomnographic (PSG) recordings;

---

[1]That is, one could understand how the prediction is made by examining similar samples.
[2]Experiments about the effect of $d$ on performance and overhead are provided in the Appendix.

PN2017 (2017 PhysioNet Challenge) (Clifford et al., 2017; Goldberger et al., 2000), a public electro-cardiogram (ECG) dataset for rhythm (particularly Atrial Fibrillation) classification. For the training set, we follow Hong et al. (2019); Jing et al. (2021) for PN2017 and IIIC, and used 69 patients' data for ISRUC. For the remaining data, 5% is used as the calibration set and 95% for testing. We perform additional experiments after splitting into training/calibration/test sets by patients for IIIC and ISRUC[3], marked as the "pat" version in tables. The calibration/test split is 20/80 in "IIIC (pat)" and "ISRUC (pat)" because the number of patients is small. For IIIC and ISRUC, we follow the standard practice and train a CNN (ResNet) on the spectrogram (Biswal et al., 2017; Ruffini et al., 2019; Yuan et al., 2019; Yang et al., 2022). For PN2017, we used a top-performing model from the 2017 PhysioNet Challenge, MINA (Hong et al., 2019).

Table 2: Accuracy in % ($\uparrow$ means higher=better). Accuracy numbers lower than the uncalibrated predictions are in dark red and the best are in **bold** (both at p=0.01). KCal typically improves or maintains the accuracy.

| Accuracy $\uparrow$ | UnCal | TS | DirCal | I-Max | Focal | Spline | IOP | GP | MMCE | KCal |
|---|---|---|---|---|---|---|---|---|---|---|
| IIIC (pat) | 58.68±1.42 | 58.68±1.42 | **63.17±1.42** | 57.20±1.32 | 54.35±1.64 | 58.51±1.32 | 58.68±1.42 | 58.68±1.42 | 58.05±1.37 | **61.67±2.22** |
| IIIC | 58.53±0.06 | 58.53±0.06 | 63.80±0.10 | 56.96±0.14 | 54.41±0.05 | 58.36±0.20 | 58.53±0.06 | 58.52±0.06 | 58.06±0.04 | **66.32±0.21** |
| ISRUC (pat) | 75.11±0.77 | 75.11±0.77 | **75.57±0.91** | 75.54±0.68 | 73.79±0.72 | 75.11±0.79 | 75.11±0.77 | 75.11±0.76 | **76.26±0.59** | 76.13±0.89 |
| ISRUC | 74.66±0.08 | 74.66±0.08 | 76.08±0.16 | 75.15±0.07 | 73.34±0.09 | 74.69±0.09 | 74.66±0.08 | 74.66±0.09 | 75.95±0.07 | **77.45±0.16** |
| PN2017 | 54.67±0.14 | 54.67±0.14 | 60.00±0.22 | 57.55±0.39 | 13.78±0.13 | 55.11±0.84 | 55.15±1.48 | 54.69±0.15 | 51.90±0.07 | **60.36±0.61** |
| C10 (ViT) | 98.94±0.05 | 98.94±0.05 | 98.94±0.05 | 98.94±0.05 | 98.76±0.06 | 98.94±0.05 | 98.94±0.05 | 98.94±0.06 | 98.93±0.07 | **98.98±0.09** |
| C10 (Mixer) | 98.17±0.08 | 98.17±0.08 | 98.03±0.09 | 98.13±0.08 | 96.98±0.08 | 98.17±0.08 | 98.17±0.08 | 98.16±0.08 | 98.15±0.06 | **98.14±0.06** |
| C100 (ViT) | 92.09±0.16 | 92.09±0.16 | 92.08±0.14 | 91.95±0.17 | 91.21±0.12 | 92.09±0.16 | 92.09±0.16 | 92.09±0.16 | **92.41±0.17** | 92.37±0.15 |
| C100 (Mixer) | 87.53±0.20 | 87.53±0.20 | 87.24±0.22 | 87.10±0.21 | 86.49±0.23 | 87.53±0.20 | 87.53±0.20 | 87.51±0.20 | **88.13±0.25** | 87.55±0.16 |
| SVHN (ViT) | 95.93±0.05 | 95.93±0.05 | 95.93±0.05 | 95.85±0.06 | 95.70±0.08 | 95.93±0.05 | 95.93±0.05 | 95.93±0.05 | **96.48±0.04** | 96.42±0.05 |
| SVHN (Mixer) | 95.85±0.04 | 95.85±0.04 | 95.98±0.04 | 95.85±0.05 | 95.24±0.04 | 95.85±0.04 | 95.85±0.04 | 95.85±0.05 | 95.58±0.05 | **96.10±0.04** |
| ImageNet | 80.44±0.24 | 80.44±0.24 | 79.55±0.24 | 80.34±0.28 | – | 80.22±0.27 | 80.44±0.24 | 80.44±0.24 | – | 79.64±0.24 |

Table 3: Class-wise ECE in $10^{-2}$ ($\downarrow$ means lower=better). The best accuracy-preserving method is in **bold** (p=0.01). The lowest but not accuracy-preserving number is underscored. KCal almost always achieves the lowest class-wise ECE, while maintaining accuracy.

| CECE $\downarrow$ | UnCal | TS | DirCal | I-Max | Focal | Spline | IOP | GP | MMCE | KCal |
|---|---|---|---|---|---|---|---|---|---|---|
| IIIC (pat) | 8.07±0.27 | 8.97±0.85 | **5.13±1.48** | 9.23±0.98 | 8.99±0.53 | 8.56±0.62 | 8.33±0.50 | 7.95±0.64 | 7.12±0.43 | **4.68±1.27** |
| IIIC | 7.96±0.02 | 8.96±0.52 | **2.24±0.13** | 8.76±0.26 | 8.78±0.02 | 8.43±0.21 | 8.01±0.25 | 7.52±0.23 | 6.70±0.25 | **2.03±0.26** |
| ISRUC (pat) | **4.48±0.24** | 4.69±0.76 | 4.18±0.90 | 8.56±1.00 | 9.23±0.21 | 4.68±0.46 | 4.60±0.60 | 4.64±0.43 | 4.08±0.36 | **3.82±1.24** |
| ISRUC | 4.49±0.02 | 5.17±0.77 | 2.71±0.40 | 9.22±0.85 | 9.05±0.03 | 4.73±0.15 | 4.67±0.36 | 4.67±0.27 | 4.10±0.22 | **1.90±0.28** |
| PN2017 | 12.17±0.07 | 12.31±0.23 | **4.30±0.47** | 9.92±1.16 | 17.31±0.09 | 8.61±0.73 | 12.09±0.34 | 12.17±0.07 | 12.35±0.39 | **4.25±1.26** |
| C10 (ViT) | 3.19±0.01 | 0.76±0.04 | 0.83±0.06 | **0.68±0.05** | 4.82±0.07 | 0.90±0.04 | 0.81±0.06 | **0.74±0.06** | 1.11±0.27 | **0.74±0.07** |
| C10 (Mixer) | 3.11±0.02 | 1.45±0.12 | 1.23±0.10 | **1.24±0.17** | 6.70±0.03 | 1.28±0.09 | 1.30±0.07 | **1.21±0.07** | 1.43±0.19 | **1.17±0.10** |
| C100 (ViT) | 5.90±0.05 | 5.27±0.20 | 4.44±0.13 | 4.96±0.17 | 5.53±0.06 | **4.41±0.14** | 4.72±0.12 | 4.65±0.16 | **4.27±0.23** | **4.32±0.10** |
| C100 (Mixer) | 5.39±0.04 | 5.82±0.17 | 5.25±0.14 | 5.79±0.24 | 5.72±0.05 | 4.92±0.18 | 5.34±0.23 | 5.09±0.15 | 5.26±0.19 | **4.62±0.10** |
| SVHN (ViT) | 3.37±0.01 | 2.31±0.56 | **1.22±0.06** | 2.64±0.20 | 5.89±0.03 | 1.34±0.05 | 1.39±0.06 | 1.40±0.05 | 1.47±0.11 | **1.23±0.10** |
| SVHN (Mixer) | 3.20±0.01 | 3.06±0.61 | **1.21±0.12** | 2.64±0.17 | 5.59±0.02 | 1.45±0.09 | 1.44±0.06 | 1.46±0.06 | 1.64±0.13 | 1.40±0.08 |
| ImageNet | 2.96±0.02 | 3.25±0.07 | 5.60±0.23 | 2.82±0.19 | – | **2.17±0.06** | 2.30±0.14 | 2.42±0.06 | – | 1.94±0.04 |

Table 4: ECE in $10^{-2}$ ($\downarrow$ means lower=better). The best accuracy-preserving method is in **bold** (p=0.01). The lowest but not accuracy-preserving number is underscored. KCal is usually on par or better than the best baseline.

| ECE $\downarrow$ | UnCal | TS | DirCal | I-Max | Focal | Spline | IOP | GP | MMCE | KCal |
|---|---|---|---|---|---|---|---|---|---|---|
| IIIC (pat) | 9.32±1.01 | **5.00±2.75** | 2.92±1.59 | 10.52±4.05 | 7.53±0.55 | **4.58±2.04** | **4.57±2.14** | **3.86±1.63** | 6.33±3.28 | **4.34±1.35** |
| IIIC | 9.28±0.03 | 4.45±1.52 | 1.39±0.19 | 10.16±0.81 | 7.25±0.05 | 3.20±0.64 | 3.50±0.41 | **1.80±0.49** | 4.78±2.24 | 2.62±0.59 |
| ISRUC (pat) | 3.59±0.32 | **2.73±1.53** | 2.97±0.97 | 8.86±1.39 | 14.88±0.43 | **1.98±0.35** | 2.45±1.36 | **2.00±0.53** | 2.12±0.93 | **2.78±1.25** |
| ISRUC | 3.46±0.06 | 3.82±1.69 | 2.27±0.69 | 9.58±1.23 | 14.70±0.06 | **1.50±0.53** | 2.71±0.96 | 2.09±0.74 | 2.12±1.03 | **1.36±0.41** |
| PN2017 | 16.70±0.22 | 16.99±0.73 | 5.64±0.75 | 10.40±1.35 | 24.63±0.13 | **6.84±2.09** | 16.07±2.03 | 16.66±0.21 | 13.49±1.07 | **4.78±1.48** |
| C10 (ViT) | 9.15±0.05 | 0.75±0.11 | 0.40±0.04 | 0.51±0.07 | 7.17±0.07 | 0.39±0.08 | 0.39±0.04 | **0.21±0.06** | 0.42±0.29 | 0.40±0.05 |
| C10 (Mixer) | 9.04±0.06 | 1.06±0.12 | 0.61±0.07 | 0.91±0.14 | 12.53±0.06 | **0.36±0.06** | 0.66±0.09 | **0.34±0.10** | 0.91±0.44 | 0.59±0.09 |
| C100 (ViT) | 11.64±0.14 | 2.77±0.46 | **0.74±0.16** | 3.28±0.42 | 9.97±0.09 | 1.08±0.18 | 1.07±0.19 | **0.88±0.11** | 1.05±0.30 | 1.50±0.32 |
| C100 (Mixer) | 13.71±0.15 | 3.03±0.34 | 1.06±0.28 | 4.75±0.27 | 14.35±0.21 | **1.25±0.29** | 1.70±0.66 | **1.08±0.26** | 1.93±0.49 | 3.07±0.49 |
| SVHN (ViT) | 10.10±0.05 | 2.43±2.72 | **0.60±0.07** | 2.05±0.18 | 12.17±0.08 | 0.74±0.10 | **0.62±0.08** | **0.64±0.07** | **0.72±0.21** | **0.64±0.12** |
| SVHN (Mixer) | 10.29±0.04 | 3.19±2.55 | **0.66±0.05** | 2.13±0.10 | 11.09±0.06 | 0.78±0.11 | **0.60±0.08** | 0.72±0.06 | 0.72±0.28 | 0.73±0.10 |
| ImageNet | 3.21±0.15 | 3.52±0.13 | 4.30±0.68 | 7.97±0.35 | – | 1.10±0.20 | 1.31±0.47 | **0.87±0.12** | – | 1.43±0.34 |

## 4.2 BASELINES METHODS

We compare KCal with the multiple state-of-the-art calibration methods, including Temperature Scaling (`TS`) (Guo et al., 2017), Dirichlet Calibration (`DirCal`) (Kull et al., 2019), Mutual-information-maximization-based Binning (`I-Max`) (Patel et al., 2021), Gaussian Process Calibration

---
[3]PN2017 did not provide patient IDs, so we cannot split by patient.

Table 5: Brier Score in $10^{-2}$ ($\downarrow$ means lower=better). The best accuracy-preserving methods are in **bold** (p=0.01). The lowest but not accuracy-preserving number is underscored.

| Brier ↓ | UnCal | TS | DirCal | I-Max | Focal | Spline | IOP | GP | MMCE | KCal |
|---|---|---|---|---|---|---|---|---|---|---|
| IIIC (pat) | 21.30±0.25 | 20.70±0.69 | **18.94±0.55** | 21.09±1.29 | 21.48±0.19 | 20.43±0.50 | 20.52±0.58 | 20.33±0.42 | 21.11±0.71 | **19.33±0.78** |
| IIIC | 21.35±0.01 | 20.62±0.27 | 18.33±0.04 | 20.83±0.19 | 21.46±0.01 | 20.21±0.09 | 20.39±0.09 | 20.05±0.08 | 20.86±0.26 | **17.54±0.10** |
| ISRUC (pat) | 15.26±0.25 | 15.20±0.31 | 15.37±0.38 | 16.25±0.49 | 18.55±0.18 | 15.11±0.26 | 15.16±0.31 | 15.16±0.29 | **14.69±0.22** | **14.97±0.29** |
| ISRUC | 15.46±0.03 | 15.50±0.19 | 15.07±0.09 | 16.62±0.33 | 18.77±0.01 | 15.31±0.05 | 15.39±0.10 | 15.35±0.06 | 14.91±0.08 | **14.28±0.08** |
| PN2017 | 26.61±0.05 | 26.74±0.27 | **22.44±0.15** | 24.58±0.59 | 17.79±0.03 | 23.28±0.37 | 26.39±0.69 | 26.61±0.05 | 26.41±0.44 | **22.56±0.28** |
| C10 (ViT) | 1.76±0.03 | 0.89±0.06 | **0.78±0.04** | 0.84±0.04 | 1.75±0.03 | **0.79±0.04** | **0.79±0.04** | 0.78±0.04 | 0.85±0.10 | **0.75±0.05** |
| C10 (Mixer) | 2.29±0.03 | 1.48±0.07 | 1.42±0.05 | 1.46±0.08 | 4.16±0.04 | 1.39±0.04 | 1.40±0.05 | **1.37±0.04** | **1.45±0.16** | **1.34±0.04** |
| C100 (ViT) | 6.94±0.08 | 5.35±0.15 | 5.17±0.10 | 5.48±0.14 | 6.93±0.07 | 5.19±0.09 | 5.18±0.10 | 5.14±0.09 | **4.81±0.10** | 5.01±0.08 |
| C100 (Mixer) | 10.15±0.11 | 7.94±0.17 | 7.82±0.12 | 8.23±0.17 | 10.91±0.08 | 7.76±0.12 | 7.82±0.15 | 7.72±0.13 | **7.38±0.16** | 7.61±0.09 |
| SVHN (ViT) | 3.99±0.03 | 3.03±0.34 | 2.78±0.04 | 2.99±0.07 | 5.03±0.03 | 2.80±0.03 | 2.79±0.04 | 2.79±0.04 | **2.43±0.02** | 2.49±0.03 |
| SVHN (Mixer) | 4.03±0.03 | 3.21±0.36 | 2.77±0.03 | 3.04±0.04 | 5.06±0.04 | 2.84±0.03 | 2.81±0.04 | 2.81±0.04 | 3.03±0.02 | **2.68±0.03** |
| ImageNet | 11.15±0.14 | 11.20±0.15 | 12.03±0.21 | 11.93±0.18 | – | **10.68±0.13** | **10.69±0.13** | 10.67±0.12 | – | 11.14±0.10 |

Table 6: Ranks for different evaluation metrics. The best rank is underscored. In general, KCal consistently outperforms baselines on Accuracy, CECE and Brier, and the difference between most methods on ECE is small.

| Ranking | UnCal | TS | DirCal | I-Max | Focal | Spline | IOP | GP | MMCE | KCal |
|---|---|---|---|---|---|---|---|---|---|---|
| ECE | 8.42±1.43 | 6.68±1.11 | 3.33±1.80 | 7.73±1.55 | 9.39±0.95 | 3.51±1.06 | 4.25±1.35 | 2.91±1.66 | 4.52±0.98 | 3.84±1.35 |
| Accuracy | 5.03±1.30 | 5.03±1.30 | 6.41±2.36 | 6.41±2.36 | 9.99±0.03 | 5.56±0.93 | 5.01±1.27 | 5.64±1.16 | 4.74±3.30 | 2.70±2.01 |
| CECE | 6.99±1.95 | 7.41±1.60 | 3.31±2.08 | 6.82±2.67 | 9.46±0.61 | 4.59±2.06 | 5.12±1.13 | 4.37±1.27 | 4.69±1.99 | 1.83±0.76 |
| Brier | 8.18±1.52 | 6.91±0.85 | 3.86±2.08 | 7.42±1.06 | 8.98±2.67 | 4.23±1.05 | 4.88±1.24 | 3.89±1.83 | 4.11±2.89 | 2.05±1.17 |
| Average | 7.16 | 6.51 | 3.76 | 7.09 | 9.46 | 4.47 | 4.81 | 4.20 | 4.51 | 2.61 |

(GP) (Wenger et al., 2020), Intra Order-preserving Calibration (IOP) (Rahimi et al., 2020), Splines-based Calibration (Spline) (Gupta et al., 2021), Focal-loss-based calibration (Focal) (Mukhoti et al., 2020), MMCE-based calibration (MMCE) (Kumar et al., 2018).

## 4.3 EVALUATION METRICS

We report standard evaluation metrics: Accuracy, class-wise expected calibration error (CECE) (Kull et al., 2019; Patel et al., 2021; Nixon et al., 2019), expected calibration error (ECE) (Guo et al., 2017), and Brier score (Brier, 1950). CECE is typically used as a proxy to evaluate full calibration quality, because directly binning basing on the entire vector $\hat{\mathbf{p}}$ requires exponentially (in $K$) many bins. Similar to Patel et al. (2021); Nixon et al. (2019), we ignore all predictions with very small probabilities (less than $\max\{0.01, \frac{1}{K}\}$). ECE, on the other hand, only measures confidence calibration (Def 2). For both ECE and CECE, we use the "adaptive" version with equal number of samples in each bin (with 20 bins), because this is shown to measure the calibration quality better than the equal-width version (Nixon et al., 2019). Brier score can be viewed as the sum of a "calibration" term, and a "refinement" term measuring how discriminative a model is (Kull & Flach, 2015). Here we focus on the brier score of the top class. We refer to (Guo et al., 2017; Kull et al., 2019; Nixon et al., 2019) for further discussion on these metrics.

## 4.4 RESULTS

The results are presented in Tables 2, 3, 4 and 5. All experiments are repeated 10 times by reshuffling calibration and test sets, and the standard deviations are reported. For ImageNet, we skipped Focal and MMCE because the base NN is given and these methods require training from scratch. Due to space constraints, we include ablation studies in the Appendix.

In general, KCal has the best CECE, accuracy and Brier score, and is highly competitive in terms of ECE as well. Note that KCal is also the only method with provable calibration guarantee. TS is effective in controlling overall ECE but shows little improvement on CECE over UnCal. DirCal often ranks high for the calibration quality but tends to decrease accuracy as $K$ increases. DirCal's performance also has a higher cost: Every experiment requires training over hundreds of models with SGD and taking the best ensemble, accounting for most of the experiment computation cost in this paper. The amount of tuning suggested for good performance indicates sensitivity to the choice of hyper-parameters, which we have indeed observed to be the case. Spline, IOP and GP are similar to DirCal on vision datasets, but generally perform worse on the healthcare datasets.

In Patel et al. (2021), `I-Max` lowers ECE and CECE significantly. However, it has a critical issue - it does *not* produce a valid probability vector[4]. Once normalized, as reported in our experiments, the performance worsens. Since calibrating all the classes simultaneously is *the* distinguishing challenge in multiclass classification, we interpret the observation as: If this normalization constraint is removed, the "optimization problem" (to lower calibration error) is much simpler, but the results are invalid hence unusable probability vectors. `Spline` also requires a re-normalization step, but its performance stays consistent. `Focal` is worse than the `UnCal` in many experiments. While calibration performance may improve by combing `Focal` with other methods, the drop in accuracy is harder to overcome[5]. We also observed that for healthcare datasets, being able to tune on a different set of patients boosts the performance significantly. This is reflected in the accuracy gain for `DirCal` and KCal, and suggests that the embeddings/logits are quite transferable, but the prediction criteria itself can vary from patient to patient.

Finally, we summarize the rankings of all datasets in Table 6. It is clear that KCal consistently improves calibration quality for all classes and maintains or improves accuracy. And if we look at only the confidence prediction (Brier or ECE), KCal is still highly competitive.

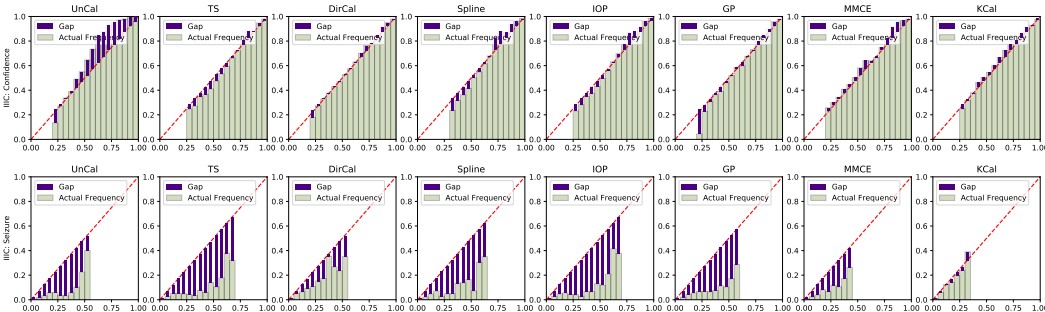

Figure 2: Reliability diagrams for the predicted class (top) and Seizure (bottom) in IIIC. All methods calibrate confidence well, but only KCal achieves reasonable calibration quality for Seizure.

## 4.5 CASE STUDY FOR SEIZURE PREDICTION

We show the reliability diagrams (Kull et al., 2019; Guo et al., 2017) on the IIIC dataset to illustrate the importance of full calibration in Figure 2. We include both the the predicted class (confidence calibration) and Seizure. More reliability diagrams can be found in the Appendix, and the results are consistent for all classes. The un-calibrated predictions have large gaps for both confidence and Seizure. Most baselines provide calibrated confidence calibration, but fail to calibrated the output for the rare class Seizure. KCal, on the other hand, achieves the most consistent results. We note again that since all competing classes must be considered together for any clinical decision, *full* calibration is indispensable in medical applications.

## 5 CONCLUSION

This paper proposed KCal, a learned-kernel-based calibration method for deep learning models. KCal consists of a supervised dimensionality reduction step on the penultimate layer neural network embedding to improve efficiency. A KDE classifier using the calibration set is employed in this new metric space. As a natural consequence of the construction, KCal provides a calibrated probability vector prediction for all classes. Unlike most existing calibration methods, KCal is also *provably* asymptotically *fully* calibrated with finite sample error bounds. We also showed that empirically, it outperforms existing state-of-the-art calibration methods in terms of accuracy and calibration quality. Moreover, KCal is more robust to distributional shift, which is common in high-risk applications such as healthcare, where calibration is far more crucial. The major limitation of KCal is the need to store the entire calibration set, which is a small overhead with the dimension reduction step and potential improvements.

---

[4]It generates a vector whose sum ranges from 0.4 to 2.0 in our experiments. The range is wider for a larger $K$.

[5]In PN2017, rare classes are oversampled during training (Hong et al., 2019). While this did not cause issues for other calibration methods, the distributional shift at test time seems catastrophic for `Focal`.

ACKNOWLEDGMENTS

This work was supported by NSF award SCH-2205289, SCH-2014438, IIS-1838042, NIH award R01 1R01NS107291-01.

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
