# OpenReview forum: "Taking a Step Back with KCal: Multi-Class Kernel-Based Calibration for Deep Neural Networks"
_ICLR.cc/2023/Conference — ICLR 2023 poster_

### Official Review · Reviewer_2JZP · 2022-10-20

**Confidence:** 3
**Correctness:** 4
**Technical Novelty And Significance:** 3
**Empirical Novelty And Significance:** 2
**Recommendation:** 6

**Clarity, Quality, Novelty And Reproducibility:**

I found the paper well written, easy to follow, clear and pedagogical.

Kcal can be summarised as fine-tuning a DNN with a KDE as last layer instead of a softmax. I have not seen such as attempt earlier and this idea is novel to me.

Note 1: To improve clarity, it could be helpful to have a diagram showing an example of train set with a batch and a background set.

Note 2: There is a mistake in Definition 2 (Confidence Calibration). The argmax should be taken over the whole vector of probabilities, and the conditioning occurs on the maximum entry of $\hat p$.

Note 3: In the sentence “Calibration is the process of closing the gap between the prediction and the ground truth distribution“, it would be better to use the word “reducing” instead of “closing” as calibration alone, even of a perfectly accurate classifier, is not enough to remove the ground truth probabilities.


**Strength And Weaknesses:**

**Strength**.
Kcal targets **full calibration**, i.e., a calibration conditioned on the full probability vector rather than just conditioned on the probability of predicted class (confidence calibration). This is an important problem since as stated by the authors, small probabilities should also be reliable in high stakes applications such as healthcare.

Kcal provides improvements in terms of Accuracy, Class-wise ECE and Brier score across most datasets.

**Weaknessess**.
Using a KDE-based estimator requires an adapted training scheme, and comes with a presumably high computational cost. What is the training time of Kcal across the different datasets?
It also introduces multiple hyper parameters: the kernel bandwidth, the projection architecture and output size, the batch size used to learn the projection, the number of samples used to build the KDE.
Therefore, it comes with both a certain computational and implementation complexity.

No insights are provided to explain the performance of Kcal compared to the other baselines. I am wondering wether it could be due to the fact that it re-uses the whole train set to fit the projection map, while other methods such as Temperature Scaling only calibrate based on the smaller calibration set.

**Question**.
This is a question regarding the choice of kernel bandwidth $b$. Since it can be folded in the projection map, it is implicitly learned while training the projection map. We see in appendix that across all datasets, the best kernel bandwidth equals 1. it would make sense if it has already been learned with the projection map. Hence the question: it is useful to have a bandwidth selection step on the calibration set?

**Summary Of The Paper:**

This paper proposes a method called Kcal for the (full) calibration of deep neural networks.
Kcal consists in replacing the last layer (usually a softmax) of a neural network with a learned projection layer for dimensionality reduction, followed by a Kernel Density (KDE)-based estimator to produce output probabilities. The learnable parameters introduced by Kcal are the parameters of the dimension-reducing projection layer $\Pi$, as well the kernel bandwidth $b$. The parameters of $\Pi$ are learned by (stochastic) gradient descent with an appropriate training strategy: contrary to classical neural networks, the KDE “layer” involves all training points and not only the point which is flowed through the network. The kernel bandwidth $b$ is chosen on a calibration set to minimise the log-loss.

A theoretical analysis is provided in the form of high probability error bound, which shows that as the per-class sample size increases, the estimated probabilities converge to the true probability of the class $Y$ given the input score, i.e, the estimated probabilities are calibrated.

Experiments are provided for 4 vision datasets (CIFAR10, CIFAR100, SVHN, ImageNet) each time with a choice of pre-trained architecture, and 3 health monitoring datasets. Kcal is then compared to 8 calibration baselines in terms of ECE (top-label and class-wise), accuracy and Brier score of the top class, and shown to provide favourable performance across all measures (or on par for ECE).

**Summary Of The Review:**

I think that the clarity of the paper as well as the favourable empirical results outweight the weaknesses.

---

> ### Author Response · Authors · 2022-11-14
> **Thank you for your comments**
>
> Thank you very much for the thorough review and suggestions. We clarify some of the details below:
>
> ## Computation cost
> The inference time is included in Appendix F and is very small compared to the base DNN.
> The training only needs to happen once, and is relatively fast as we are only tuning a very shallow MLP with few parameters (it converged in 4 hours on ImageNet on a desktop GPU).
>
> ## Additional hyperparameters
> The kernel bandwidth can be automatically selected.
> We provide an ablation study for the projection architecture (linear is very good already) as well as for the output size. While it is true that one needs to choose the batch size in training the projection, we found that they did not seem to matter much in our experiments, so we just fixed them without tuning.
> The fact that these hyperparameters are fixed for all experiments (across a variety of datasets, including ones in healthcare), along with our ablation studies, suggests that KCal is not very sensitive to their values.
> Moreover, many baseline calibration methods also have such hyperparameters to select and can require extensive tuning, so we think we can conclude that KCal is no more complex compared with our baselines.
>
>
> ## Source of performance gain
> > No insights are provided to explain the performance of Kcal compared to the other baselines
>
> We discuss major differences with other methods in Section 3.5. In our setting, performance has two aspects - accuracy preservation and calibration error.
> For accuracy preservation, as discussed in the first paragraph of Section 3.5, we think the difference boils down to whether a nonparametric KDE mapping or a linear mapping (from the last layer to the prediction) is more expressive.
> We would expect that, in general, a linear map is better for fewer data points but the nonparametric map is better with more data (see Appendix I).
> As for calibration quality, we believe the fact that KCal does not require an extra normalization step is important (3rd paragraph of Section 3.5).
>
> > I am wondering whether it could be due to the fact that it re-uses the whole train set to fit the projection map
>
> The training of the projection map involves transforming embeddings that more suitable for classification with a linear head, to embeddings more suitable for similarity comparisons.
> The prediction step still uses the same calibration set as all baselines.
> To see our point, in Appendix I, with very small calibration sets, we do observe a deterioration in performance for KCal.
> We chose to train $\mathbf{\Pi}$ on the training set for the theoretical guarantee (as opposed to training it on the calibration set).
>
> ## Question: importance of bandwidth selection?
> Thank you for the question.
> We think this step is necessary,
> In Appendix G (Figure 8), we show the relationship between the empirically chosen bandwidth vs. the optimal bandwidth computed from $m$, and we see a close-to-linear relationship in many cases.
> Morever, the best bandwidth is usually not $1$ (except for CIFAR10-ViT).
> Although the purpose of this section is to show that one could update the bandwidth in an online fashion, it also shows that the optimal bandwidth is generally a function of sample size.
>
> ## Notes:
> Thank you for the various suggestions/notes.
>
> > it could be helpful to have a diagram showing an example of train set with a batch and a background set.
>
> Could you elaborate what you mean by this?
> We are space-constrained in the main text, but could add some remarks or additional experiments to the Appendix.
>
> >  The argmax should be taken over the whole vector of probabilities, and the conditioning occurs on the maximum entry of $\hat{p}$
>
> We used a slightly different notation because of the $k$ under the argmax, so the argmax does not take the whole vector.
> We have fixed the typo regarding the conditioning of $\hat{p}$.
>
> > it would be better to use the word “reducing” instead of “closing” as calibration alone
>
> To clarify, by "ground truth distribution", we refer to the distribution given a specific prediction. This gap could be "closed" (in expectation). We have added a note on this in our draft.

---

### Official Review · Reviewer_W8Nj · 2022-10-24

**Confidence:** 5
**Correctness:** 3
**Technical Novelty And Significance:** 3
**Empirical Novelty And Significance:** 2
**Recommendation:** 6

**Clarity, Quality, Novelty And Reproducibility:**

Clarity&quality: This paper is clearly written.
Novelty: kernel is not new to the probability calibration but accroading to the authors, kde is firstly introduced.
Reproducibility: The code to replicate all our experimental results is submitted along with supplementary materials.

**Strength And Weaknesses:**

Strengths:
1. A well-established method that builds a kernel-based calibration method.
2. Theoretical proof of KCal.
3. Experimental result analysis.

Weakness:
1. Somewhat bold claims of first theoretical guaratee/kde calibration method: The conformal predictions is well-studied with theoretical guarantee[Shafer 2008, Angelopoulos 2020].  The kernel mean embedding for uncertainty calibration [Kumar 2008, Cui 2020, Bhatt 2022].
2. Calibration performance not better in all cases
3. Computation time not fully explained/demostrated. As said in the paper, computation overhead of KCal is generally larger than ordinary methods and this needs to be empirically analyzed in different scales and acceleration settings.


References
1. Shafer G, Vovk V. A Tutorial on Conformal Prediction[J]. Journal of Machine Learning Research, 2008, 9(3).
2. Angelopoulos A, Bates S, Malik J, et al. Uncertainty sets for image classifiers using conformal prediction[J]. arXiv preprint arXiv:2009.14193, 2020.
3. Kumar A, Sarawagi S, Jain U. Trainable calibration measures for neural networks from kernel mean embeddings[C]//International Conference on Machine Learning. PMLR, 2018: 2805-2814.
4. Cui P, Hu W, Zhu J. Calibrated reliable regression using maximum mean discrepancy[J]. Advances in Neural Information Processing Systems, 2020, 33: 17164-17175.
5. Bhatt D, Mani K, Bansal D, et al. f-Cal: Aleatoric uncertainty quantification for robot perception via calibrated neural regression[C]//2022 International Conference on Robotics and Automation (ICRA). IEEE, 2022: 6533-6539.


**Summary Of The Paper:**

This paper proposes a new Kernel-based calibration method called KCal which  learns a metric space on the penultimate-layer
latent embedding and generates predictions using kernel density estimates on a calibration set. Theoretical and empirical results shows the effeciveness of KCal.

**Summary Of The Review:**

This paper proposes KCal which is claimed as the first kde-based calibration method that brings theoretical analysis. The method itself is interesting and promising but some claims and empirical results needs improving.

---

> ### Author Response · Authors · 2022-11-14
> **Clarifications and point-by-point response**
>
> Thank you very much for the constructive feedback on our work! We would like to make a few clarifications that we hope will resolve the questions you raise.
>
> ## Theoretical Guarantee
> Thank you for pointing out several works on conformal prediction and regression tasks. We are aware of these works and would like to take this opportunity to highlight how our guarantees differ.
>
> ### Calibration in the suggested references
> To begin, we would like to note that *calibration* is an overloaded term and is used in many contexts. At a high level, we see the similarity in the works you mentioned, in that all of us are trying to calibrate a distribution. However, this is where the similarity ends. Conformal prediction is a powerful methodology used widely to provide provable coverage guarantees for prediction sets or prediction intervals.
> The type of guarantee they provide - *coverage guarantees* (e.g., Theorem/proposition 1 in [Angelopoulos 2020]) for the prediction sets in [Shafer 2008, Angelopoulos 2020] - is very different from ours. [Cui 2020, Bhatt 2022] seem related because they discuss the calibration of the distribution - theorem 1 in [Cui 2020] is about the convergence of the p-value. However, they focus on *regression*, but our paper is related only to classification. In fact, Section C.2 of [Cui 2020] explicitly distinguishes itself from our setting. (Although [Cui 2020] could probably be extended to binary classification, the case of multi-class classification is more difficult.) In a sense, all of these works are trying to provide a *marginally* valid p-value. (Recently, people started investigating various types of approximate conditional validity in conformal prediction, but this is beyond the scope of this discussion.)
>
> ### Calibration in our paper
> We are also aware that "calibration set" is also a term frequently used in the conformal prediction literature, which can cause some confusion as we also have a calibration set.
> We focus solely on the calibration of (DNN) classifiers, following the line of work of (Guo et al., 2017).
> This is an important area with many papers (as reflected by the number of baseline methods we compared with).
> One could potentially think of this as some sort of "conditional" calibration.
> Our calibration guarantee is on the output of the classifier (see Eq.(11)) and there is no prediction set/interval involved.
> Thus, works like [Angelopoulos 2020] could be applied *after* all the calibration baselines mentioned in our paper, of course including KCal.
> To be specific, the guarantee that we provide is on the *calibration error* (the "conditional" part is tackled using recent results on the uniform convergence of KDE).
> As noted in the paper, to the best of our knowledge KCal is the first method with *full* calibration guarantee in this sense.
>
> Finally, [Kumar 2008] does share the same task as our paper and uses a kernel in calibrating the classifier.
> However, they use the kernel to regularize the training, and do not provide calibration error bounds.
> The most similar theoretical result there is linking MMCE to ECE.
> We also compare KCal with [Kumar 2008] in our experiments.
>
> >  bold claims of first theoretical guaratee/kde calibration method
>
> We hope the clarification above could resolve this concern. However, we would be grateful if you could point us to specific points/claims that we made that gave the appearance of a bold claim. We will try our best to ensure that we are more clear about this.
>
> To summarize our response: KCal is the first calibration method that uses KDE to achieve full calibration, and to the best of our knowledge also the first calibration method that provides *full* calibration guarantee (not just for confidence calibration). We also restrict our discussion to the calibration of DNNs (although some of our theoretical results are more general).
>
> ## Calibration Performance
>
> Thanks for the suggestion. KCal is indeed not the best for all metrics. However, it is the first method with a provable full calibration guarantee, which we believe is also an important consideration for calibration (especially where full calibration is critical, such as applications in differential diagnosis). Also, despite the fact that most calibration methods are quite similar in performance, we observe that KCal seems to be the best on average (Table 6), and only slightly behind some other state-of-the-art methods in terms of one metric out of the many that we consider.
>
>
> ## Computation Time
>
> Thank you for pointing out the concern about computation time.
> We do provide such an analysis in Figure 7 (Appendix F), where we compare the effect of different $d$ (the embedding size). In general, we report that the overhead is negligible compared to the computation cost by the base DNN.

---

> > ### Comment · Reviewer_W8Nj · 2022-11-17
> > **thanks for the comment**
> >
> > Thanks for your detailed comments that resolves my concerns. I will raise my score accroadingly.

---

### Official Review · Reviewer_VY8v · 2022-10-24

**Confidence:** 4
**Correctness:** 4
**Technical Novelty And Significance:** 4
**Empirical Novelty And Significance:** 4
**Recommendation:** 8

**Clarity, Quality, Novelty And Reproducibility:**

The paper is clearly written, original, and of high quality.


**Strength And Weaknesses:**

Strengths:
* The paper is well-written and clear;
* The method is novel, theoretically justified and improves the state-of-the-art;
* The experiments have been carried out with great care in optimizing the hyperparameters for the existing methods;

Weaknesses:
* I didn't find any major weaknesses.

Minor weaknesses:
* The list of measures includes ECE, Accuracy, CECE, and Brier score. However, cross-entropy (NLL) could have been considered as well, because all proper losses decompose into refinement and calibration losses, not just the Brier score. NLL has different properties than the Brier score, and pays more attention to the probabilities near the extremes.
* The number of instances per bin could have been shown for Figure 2 (and for similar figures in the appendix), for example as a separate figure. This would help one to visually assess roughly how high ECE would be.


**Summary Of The Paper:**

The paper proposes a kernel-based calibration KCal where class proabilities are determined by per-class kernel density estimates in an embedding space. The embedding space is obtained by a multi-layer perceptron applied on top of the penultimate layer of the neural network. Extensive experiments demonstrate that KCal outperforms the methods on most relevant measures and on many datasets. KCal has asymptotic guarantees and finite sample bounds.


**Summary Of The Review:**

The paper is well-written and has very few minor weaknesses. The proposed method is performing better than existing methods for most evaluation measures and datasets. Training of the projection MLP is an extra step that has to be done on the validation data.

---

> ### Author Response · Authors · 2022-11-14
> **Thank you and response**
>
> We appreciate your positive comments on our paper. We are also thankful for your suggestions, and have updated our draft accordingly:
>
> ### NLL loss
> We have added a table to Appendix C.4. We observe that the results seem quite consistent with Brier Score.
>
> ### Figure 2
> We have added a secondary axis to the reliability diagrams in Appendix C.5 to reflect the size of the bins.
>
> Please do let us know if you would have more questions.

---

### Official Review · Reviewer_LaNX · 2022-10-28

**Confidence:** 3
**Correctness:** 4
**Technical Novelty And Significance:** 3
**Empirical Novelty And Significance:** 3
**Recommendation:** 8

**Clarity, Quality, Novelty And Reproducibility:**

- Clarity: The paper is quite clear in most places. The experimental results could perhaps be explained in a bit more detail (for example, reviewing the definitions of the various calibration scores).
- Quality: The work appears to be high-quality throughout.
- Novelty: The method seems novel. However, I am not very up-to-date with the latest calibration methods, and thus this is hard for me to evaluate.
- Reproducibility: Code is provided in the supplementary material.


**Strength And Weaknesses:**

Strengths
- The method is relatively simple, well-motivated, and easy to understand. Using KDE for calibration seems like a natural idea, given the connection between KDE and the Bayes classifier mentioned at the end of section 3.1 (on that note, it’s surprising to me that KDE has not been proposed elsewhere to generate calibrated predictions, as is claimed at the end of section 2).
- The method performs well empirically, across a wide range of experiments, and compared to a wide range of baselines.
- The method is theoretically tractable, and provably leads to calibrated predictions under certain conditions/assumptions.

Weaknesses
- Although I didn’t study the proofs in detail, I was curious whether there was anything in the proofs that was specific to the proposed learning algorithm. In particular, would the results hold for any feature space? Or is there something specific about the learned low-dimensional feature space that allows these theorems to hold? Are these standard results based on using KDE on a fixed feature space?
- As far as I can tell, no ablations are performed to better understand what aspects of the algorithm are crucial for attaining good calibrated predictions. For example, what happens if KDE is applied directly on the final-hidden-layer outputs of the DNN?
- I was also curious whether theorems 3.3 and 3.4 gave tight or loose bounds, given the real values of m, B, C, K, d, and alpha on the real tasks (can all of these be estimated?).


**Summary Of The Paper:**

This paper tackles the question of how to generate “fully calibrated” predictions in multi-class classification problems. It proposes a method, “KCal”, which learns a low-dimensional projection of a DNN’s final-hidden-layer embeddings, and then uses kernel density estimation (KDE) in this projected space to generate the fully-calibrated probability vectors.

Theoretically, this paper shows that as the size of the calibration set increases (where size here is measured in terms of the number of occurrences of the rarest class), the probability vectors generated by KCal get closer to being fully calibrated, and in the limit m -> infinity, they are perfectly calibrated.

Empirically, this paper compares KCal to a large number of other calibration methods, on a variety of computer vision and health monitoring datasets. It shows that KCal generally maintains or improves the accuracy of the uncalibrated model, while generally attaining better calibration scores relative to the baseline methods.


**Summary Of The Review:**

Overall, I believe the proposed KCal method is a novel and promising approach for generating more calibrated predictions for multi-class classification problems, and thus that this paper should be accepted.  However, my review is only medium confidence, given that this is not an area of research I have worked directly in.

---

> ### Author Response · Authors · 2022-11-14
> **Clarifications and response**
>
> We thank the reviewer for the appreciation of our work and the constructive feedback. To further clarify some aspects of our contribution, we answer the questions raised below:
>
> ### Proofs
>
> > would the results hold for any feature space?
>
> Indeed, the results would hold if the assumptions underlying our method held. The guarantees are not specific to DNN embeddings. However, we must also note that some feature space might not be practically useful -- it might not be discriminative enough -- or the constants in the theoretical results could limit its utility. This is one reason why we provide extensive empirical support to back up our claim about the effectiveness of KCal.
>
> > Are these standard results based on using KDE on a fixed feature space?
>
> Lemmas 3.1-3.2, which are used to build up toward our main theoretical results are minor adaptations to standard results. However, theorems 3.3 and 3.4 are new results. These results use some proof techniques from [Jiang 2017], but involve new adaptations, and strengthen the results in [Jiang 2017].
>
> ### Ablation Study
>
> We do report ablation studies in Appendix D, Appendix E, Appendix F, and Appendix I (please refer to the newly uploaded draft as the numbering of the appendices changed). We moved these studies to the appendices due to space constraints.
> We added a sentence at the beginning of Section 4.4 to point to them better.
>
> > what happens if KDE is applied directly on the final-hidden-layer outputs of the DNN?
>
> We have added comparisons using logits to Appendix E. As we hypothesized, it is not as good as using the last layer embedding.
>
> ### Tightness of 3.3/3.4
> Thank you for the great question. Unfortunately, we do not have a definite answer for the tightness of the bounds. We think this could be an interesting future research direction.  To give an idea about some of the nuances: The tightness of Theorem 4 depends on the tightness of Theorem 2 in [Jiang 2017]. In general, estimating the precise coefficients in the standard bounds for KDE (Lemmas 3.1 and 3.2) is hard. In Appendix G we try to confirm that in practice, the optimal bandwidth in Lemma 3.2, instead of each of the coefficients, is roughly estimable. This difficulty unavoidably propagates to Theorem 3.3 and 3.4. In fact, the *measurement* of the calibration error (the left-hand-side of what's inside $\mathbb{P}$ in Eq.(11)) itself is very difficult and attracts quite some research. We think it might be beyond the scope of this paper to estimate these coefficients.

---

### Comment · Reviewer_VY8v · 2022-11-14
**Similarity to Zhang et al 2020 to be clarified**

The area chair raised the question about what exactly are the relationships of this work with the work of Zhang et al 2020 which also involves KDE in the context of calibration. Currently, the paper refers to Zhang et al 2020 but does not elaborate on the exact relationship of the KDE part in that work. Furthermore, the paper currently claims that "Interestingly, no one has used kernel density estimation (KDE) to generate calibrated predictions to the best of our knowledge". Taken literally, I think the claim is valid because Zhang et al 2020 used KDE for evaluating calibration and not for generating calibrated predictions.  But then the question is whether the proposal is a straightforward adaptation of the work of Zhang et al from evaluation of calibration to generation of calibrated predictions. In estimating ECE, Zhang et al. define the quantity $\hat{\pi}(z)$, which is essentially an estimated calibration map that could also be used for generating calibrated predictions. A crucial difference, though, is that the current paper performs KDE in a different and learned space rather than on the classifier's outputs directly. However, I think it would be better if the authors would explicitly acknowledge in the paper that KDE has been used before in a similar situation and explain the differences from the work of Zhang et al. 2020.

Jize Zhang, Bhavya Kailkhura, and T. Yong-Jin Han. Mix-n-match : Ensemble and compositional methods for uncertainty calibration in deep learning. In Hal Daumé III and Aarti Singh (eds.), Proceedings of the 37th International Conference on Machine Learning, volume 119 of Proceedings of Machine Learning Research, pp. 11117–11128. PMLR, 13–18 Jul 2020. URL https://proceedings.mlr.press/v119/zhang20k.html.

---

> ### Author Response · Authors · 2022-11-14
> **Thank you for the question**
>
>
> Thank you for your pointed comments on [Zhang et al 2020]. We agree that the paper will benefit with some additional discussion. We have added comments at the end of section 2, while emphasizing [Zhang et al 2020] more. Please let us know if we could provide any additional clarifications in the main paper.
>
> However, we would also like to point out some major differences below.
>
> ## Construction
>
> As the reviewer correctly noted, we used a learned kernel while [Zhang et al 2020] does not.
> Due to the curse of dimensionality, directly using a fixed kernel on tasks like CIFAR-100 or even ImageNet does not sound reasonable.
> Even with a smaller number of classes, a supervised metric space could improve the calibration map given the same calibration set.
> In our experiments, without a trained kernel, the ECE is magnitudes higher than any baseline, and on CIFAR-100, for example, the accuracy is 20\% lower.
> For the purpose of [Zhang et al 2020] this is not an issue, as, like the reviewer pointed out, [Zhang et al 2020] uses this $\tilde{\pi}$ for *evaluation* purposes,  but for us the quality of the KDE calibration map itself is crucial.
> Moreover, they directly use predicted probability vector as the input to the kernel, but we proposed to use the latent embeddings, which contain more useful information according to our ablation study.
>
> ## Theoretical guarantee
>
> Another major difference is the type of theoretical guarantee we provide.
> [Zhang et al 2020] proves that the KDE-ECE evaluation metric is consistent, which could be seen as a "marginal" guarantee.
> What they proved is a result of the point-wise convergence of the KDE estimator, which is similar to our Theorem 3.3.
> However, point-wise convergence does not translate to bounds on the calibration error.
> This is actually one of the the main challenges we ran into in establishing the theoretical results for KCal.
> Our theorem 3.4 eventually provides such bounds using recent results on the uniform convergence of KDE.

---

### Decision · Program_Chairs · 2023-01-20

**Decision:**

Accept: poster

**Justification For Why Not Higher Score:**

The submission is an interesting combination of a learned latent space and a classical KDE regression method.  There is an appropriate theoretical analysis of the method showing calibration, but the algorithm does not have a high degree of novelty.  It is not well presented in the context of existing methods, with the authors strongly overstating their claims to being first.

**Justification For Why Not Lower Score:**

The paper received unanimous accept reviews.  The method uses an interesting classical method in a new context to good effect.

**Metareview: Summary, Strengths And Weaknesses:**

The submission proposes a density estimate approach to generating calibrated outputs.  It essentially combines a learning procedure for a latent space with a classic density estimate regression approach - the main predictive equation of the paper appears e.g. after (12.3) in http://faculty.washington.edu/yenchic/18W_425/Lec12_class.pdf
The main algorithmic contributions is to do this on a space learned by a neural network (Alg 1).  The reviewers were generally pleased with this contribution, with a unanimous opinion that the submission passes the threshold for acceptance.  Sometimes a simple approach based on a classic result is a valuable contribution, especially when it has nice calibration guarantees and competitive empirical results with other more recently introduced methods.

I have serious concerns about the claims of novelty in the paper, and would like that the authors remove one of their claims, namely " To the best of our knowledge, this is the first method with a full calibration guarantee."  The method essentially performs a known classical density estimation approach albeit on a learned space.  The theoretical analysis in Section 3.3 is nice, but it essentially proves that a classical method also has this guarantee.  Thus, it is highly misleading to say that this is the first method to have a calibration guarantee.  Rather, it provides an analysis showing that regression based on KDE is a good method when calibration is required, and that it is particularly effective when combined with a learned subspace and careful analysis of the setting of the kernel bandwidth.  With this improvement, the paper can be a nice contribution to ICLR.

Furthermore, there are interesting concurrent works on KDE and calibration, with the following papers using KDE for calibration appearing in NeurIPS 2022:
https://openreview.net/pdf?id=PikKk2lF6P
https://openreview.net/pdf?id=HMs5pxZq1If - it appears an earlier version of this paper is on openreview as a submission to ICLR last year
https://openreview.net/pdf?id=q85GV4aSpt


**Note From Pc:**

if the above contains the word "oral" or "spotlight" please see: "oral" presentation means -> notable-top-5% and "spotlight" means -> notable-top-25%. As stated in our emails, we are disassociating presentation type from AC recommendations

---

> ### Author Response · Authors · 2023-01-22
> **Thank you + Clarifications + Changes**
>
> Thank you very much for the recommendation and your appreciative comments about our work. We acknowledge that your comment underlines that our paper could benefit from further clarification, and we will of course incorporate your comments to improve our manuscript for the camera-ready version.
>
> Nevertheless, we would like to take this opportunity to clarify a few points that have been raised, and respectfully disagree that we have overstated our contributions. We summarize some of these clarifications and also the changes that we will make below.
>
> First and foremost, while it is obviously true that the KDE is a classical (textbook) method and that it is at the core of our procedure, that it can be used for full calibration is certainly a new contribution. Previous methods don't give a full calibration guarantee. We have been unable to find a method that satisfies all the desiderata that we specify in our abstract (particularly in the case of neural networks). Our usage of "first" was only in these contexts, as the title suggests. However, in light of your comment, we will modify the third bullet point in the summary of contributions on page 2, which we believe is the only place that could cause an appearance of misrepresentation.
>
> Next, like in our response to the previous comment from the AC (forwarded by reviewer VY8v), the ``classical result'' is only for point-wise convergence (our Theorem 3.3). This applies to the NeurIPS 2022 paper you linked (stated in Proposition 3.2), as well as the KDE-related papers mentioned to us during the review process. We would like to be very clear that theorem 3.3, however, does *not* imply full calibration. The full calibration result (Theorem 3.4) is a *new* result building upon uniform convergence in [Jiang 2017]. The difference is that the definition of full calibration conditions on $\hat{p}(X)$ instead of $X$.
>
> > it essentially proves that a classical method also has this guarantee
>
> Yes, but the full calibration guarantee is not a ``classical'' result. The point-wise convergence in Theorem 3.3, however, is. We would love to include any reference suggesting this is not the *first* full calibration guarantee being shown for KDE classification (not restricted to DL). We will, however, change the bullet points, as earlier to reflect this suggestion i.e. that we present a new analysis that shows using KDE-based methods as we do provides full calibration.
>
> > regression based on KDE is a good method
>
> Could the PCs clarify what does regression mean here? The only regression mentioned in our paper is isotonic regression. The density estimate in KCal is implicit so there is no explicit regression.